# Micromechanics Modeling of Transverse Tensile Strength for Unidirectional CFRP Composite

**DOI:** 10.3390/ma15238577

**Published:** 2022-12-01

**Authors:** Liangbao Liu, Xiaohui Zhang, Zibiao Wang, Yana Wang, Jiangzhen Guo

**Affiliations:** 1Research Institute of Aero-Engine, Beihang University, Beijing 100191, China; 2School of Engineering Medicine, Beijing Advanced Innovation Center for Biomedical Engineering, Beihang University, Beijing 100191, China; 3State Key Laboratory of Nonlinear Mechanics, Institute of Mechanics, Chinese Academy of Sciences, Beijing 100190, China; 4Surface Engineering-Division, AECC Beijing Institute of Aeronautical Materials, Beijing 100095, China; 5Key Laboratory of Advanced Composites, AECC Beijing Institute of Aeronautical Materials, Beijing 100095, China

**Keywords:** unidirectional fiber-reinforced composites, transverse strength prediction, micromechanics modeling, stress concentration factor, stress field in matrix

## Abstract

Transverse tensile strength of unidirectional (UD) composites plays a key role in overall failure of fiber-reinforced composites. To predict this strength by micromechanics, calculation of actual stress in constituent matrix is essentially required. However, traditional micromechanics models can only give the volume-averaged homogenized stress rather than an actual one for a matrix, which in practice will cause large errors. In this paper, considering the effect of stress concentration on a matrix, a novel micromechanics method was proposed to give an accurate calculation of the actual stress in the matrix for UD composite under transverse tension. A stress concentration factor for a matrix in transverse tensile direction is defined, using line-averaged pointwise stress (obtained from concentric cylinder assemblage model) divided by the homogenized quantity (obtained from a bridging model). The actual stress in matrix is then determined using applied external stress multiplied by the factor. Experimental validation on six UD carbon fiber-reinforced polymer (CFRP) specimens indicates that the predicted transverse tensile strength by the proposed method presents a minor deviation with an averaged relative error of 5.45% and thus is reasonable, contrary to the traditional method with an averaged relative error of 207.27%. Furthermore, the morphology of fracture section of the specimens was studied by scanning electron microscopy (SEM). It was observed that different scaled cracks appeared within the matrix, indicating that failure of a UD composite under transverse tension is mainly governed by matrix failure. Based on the proposed approach, the transverse tensile strength of a UD composite can be accurately predicted.

## 1. Introduction

Fiber-reinforced polymer (FRP) composites have many advantages over conventional metals or concretes, e.g., lightweight, low maintenance, excellent resistance to stress, corrosion, and impact. As a result, they are used in a variety of industries, including aerospace, construction, automotive, defense, and more. For instance, glass fiber-reinforced polymer (GFRP) composites, which are much cheaper than other FRPs, are commonly used in civil engineering construction [1]. Carbon fiber-reinforced polymer (CFRP) composites are widely used in the aerospace industry. Airbus and Boeing, the world’s two largest airliner manufactures, adopt CFRP for the latest models, knowing that the ratios of CFRP in A350 and B787 are both about 50% [2].

To use FRP composites safely and to improve the design efficiency of laminated composites, the evaluation of its strength is necessary. However, prediction of the ultimate strength of a fibrous composite still remains a great challenge [3] and is a topic that attracts a lot of attention [4,5]. As most FRP composite structures are made from a group of stacked unidirectional (UD) composites, the evaluation of the strength of UD composites is fundamental for the strength evaluation of FRP composites. Therefore, an accurate prediction of the strength properties of UD composites is of great research interest and will be the focus of this work. 

FRP composites are treated as homogeneous and anisotropic in macromechanics failure theories, including maximum stress failure theory, maximum strain failure theory, Tsai-Hill failure theory [6] and Tsai-Wu failure theory [7], etc. Although such theories have been employed with considerable success, they are phenomenological [8] and incapable of investigating the failure mechanisms and the damage mode of composites. Additionally, the macro models are anisotropic, and are composed of various strength parameters of composites in multiple directions, which require costly and time-consuming tests on the composites. 

A micromechanics model of composites is developed to estimate the strength properties of composites only based on established constituent material properties and their geometrical parameters, without requiring a large quantity of experiments on the composites [3]. Micromechanics assumes that once the fiber or the matrix is estimated to fail, the composite will lose the capacity of load bearing. This theory shows advantages over macromechanics because of, for instance, the ability to output the internal stress in fiber/matrix and identify the failure mode of a composite. In addition, some researchers have conducted multi-scale computational analysis based on micromechanics modeling and molecular dynamics (MD) simulations to investigate the microscopic failure mechanisms of unidirectional UD CFRP composites [9].

In terms of ultimate strength, a UD composite appears to be weaker in transverse direction than in longitudinal direction, in which case matrix failure is the controlling factor. This could be supported by the fact that the overall failure of UD composites usually initiates from transverse cracking [10,11]. Furthermore, given that a matrix’s compressive strength is stronger than its tensile strength, transverse tensile strength will be the weakest point that determines the lower limit of the overall strength in a UD composite. Therefore, an accurate evaluation of transverse tensile strength is essential for the failure study of UD composites. 

Using present micromechanics models, the transverse tensile strength of a UD composite is predicted to be greater than that of a pure matrix. However, experiments report opposite results [12,13,14]. The missing link is the stress concentration in the matrix induced by the process of incorporating fibers into composites: present micromechanical models only calculate internal homogenized stress in fiber and matrix without considering the stress concentration [15,16], leading to lower predictions of the stress in matrix compared to the actual stress. A more accurate micromechanical failure model should take the actual stress in matrix as input rather than the homogenized one.

The actual stress in matrix could be derived from its homogenized value multiplied by the stress concentration factor, yet the determination of the latter parameter is a complex issue. In terms of the stress concentration factor, the traditional method in mechanics of fracture defines it as the ratio of the maximum pointwise stress to the nominal stress [17]. However, for determination of actual stress in matrix using a micromechanics model, the stress concentration factor cannot be simply defined as that. Otherwise, the value calculated would be infinitely large and the matrix strength would be almost zero when there is a crack or flaw in the matrix. 

To solve this problem, some research proposed a candidate approach to define the stress concentration factor using line-averaged stress divided by a volume-averaged quantity [15]. Huang and Xin [18] concluded from experiments and analyzed that the stress-averaged line should be perpendicular to the failure surface of the composite. Pinho [19] and Gonzalez and LLorca [20] reported that, for a UD composite under transverse tension, the failure surface is perpendicular to the loading direction, thus the stress-averaged line is parallel to the loading direction. 

In summary, ignoring the effect of stress concentration on matrix and only using the homogenized stress will cause the predicted transverse tensile strength of UD composites to be much higher than measured data. On the other hand, if considering the actual stress by definition of stress concentration factor in mechanics of fracture, the predicted transverse tensile strength of UD composites will be infinitely small. Therefore, more research studies are required to investigate the actual stress in matrix by an appropriate definition of stress concentration factor. In the present study, an effective solution was proposed to address the above issue related to transverse tensile strength prediction for UD composites. Specifically, the following work was conducted: (a) evaluating the homogenized stress in fiber and matrix using a bridging model for a given transverse tensile load applied on the UD composite, (b) investigating the stress concentration factor for the matrix and obtaining the actual stress, (c) introducing efficient failure criteria to determine failure status of the constituent fiber and matrix under the actual stress, and (d) carrying out transverse tensile loading experiment and capturing scanning electron microscopy (SEM) images on a UD composite to validate the theoretical model.

## 2. Materials and Methodology

### 2.1. Determination of Homogenized Stress in the Fiber and Matrix

Suppose that the fiber and the matrix in a composite are bonded perfectly, and that any composite is heterogeneous. Then the homogenized stress *σ_i_* in a composite is given by averaging stress with respect to its representative volume element (RVE) *V*′ [14]
(1)$$\sigma_{i}=\frac{\int_{V^{\prime }}\tilde{\sigma_{i}}\;dV}{V^{\prime }}$$
where a stress with ‘∼’ on head represents a pointwise quantity. If only fiber and matrix are contained in *V*′, the above integration can be written as
(2)$$\left\{\sigma_{i}\right\}=V_{f}\left\{\sigma_{i}^{f}\right\}+V_{m}\left\{\sigma_{i}^{m}\right\}$$
where {*σ_i_*} is the stress vector of a composite, *V*_f_ and *V*_m_ are the volume fraction of fiber and matrix, and $\left\{\sigma_{i}^{f}\right\}$ and $\left\{\sigma_{i}^{m}\right\}$ are the homogenized stress vector in fiber and matrix respectively.

When both fiber and matrix are in elastic deformation and there are no thermal residual stresses, the internal homogenized stresses in fiber and matrix are correlated with each other by a bridging tensor [*A_ij_*]
(3)$$\left\{\sigma_{i}^{m}\right\}=\left[A_{ij}\right]\left\{\sigma_{j}^{f}\right\}$$

Combining Equations (2) and (3), $\left\{\sigma_{i}^{f}\right\}$ and $\left\{\sigma_{i}^{m}\right\}$ can be expressed as
(4)$$\left\{\sigma_{i}^{f}\right\}={\left(V_{f}\left[I\right]+V_{m}\left[A_{ij}\right]\right)}^{-1}\left\{\sigma_{j}\right\}$$
(5)$$\left\{\sigma_{i}^{m}\right\}=\left[A_{ij}\right]{\left(V_{f}\left[I\right]+V_{m}\left[A_{ij}\right]\right)}^{-1}\left\{\sigma_{j}\right\}$$

So far, the major differences between existing micromechanics models consist in the deduction of the bridging tensor [*A_ij_*]. By comparing predictions from 12 most well-known micromechanics models with the measurements on the elastic properties of several UD composites used in three worldwide failure exercises (WWFEs) [12,13,14], the bridging model exhibits the best accuracy in the evaluation of internal stresses for a composite overall. In this work, a bridging model is used to determine the homogenized stress in matrix and fiber for a UD composite under external loads. Explicit expressions for the bridging tensor [*A_ij_*] in Equations (4) and (5) are denoted by [21].
(6)$$\left[A_{ij}\right]=\left[\begin{array}{cccccc} A_{11} & A_{12} & A_{13} & 0 & 0 & 0\\ 0 & A_{22} & 0 & 0 & 0 & 0\\ 0 & 0 & A_{33} & 0 & 0 & 0\\ 0 & 0 & 0 & A_{44} & 0 & 0\\ 0 & 0 & 0 & 0 & A_{55} & 0\\ 0 & 0 & 0 & 0 & 0 & A_{66} \end{array}\right]$$
(7)$$A_{11}=\frac{E^{m}}{E_{11}^{f}}$$
(8)$$A_{22}=A_{33}=A_{44}=\beta +\left(1-\beta \right)\frac{E^{m}}{E_{22}^{f}},\;0<\beta <1$$
(9)$$A_{55}=A_{66}=\alpha +\left(1-\alpha \right)\frac{G^{m}}{G_{12}^{f}},\;0<\alpha <1$$
(10)$$A_{12}=A_{13}=\frac{\upsilon^{m}E_{11}^{f}-E^{m}\upsilon_{12}^{f}}{E^{m}-E_{11}^{f}}\left(A_{11}-A_{22}\right)$$
where $E_{11}^{f}$, $G_{12}^{f}$, and $\nu_{12}^{f}$ are Young’s modulus, shear modulus, and Poisson’s ratio of the fiber in a longitudinal plane, $E_{22}^{f}$ is Young’s modulus of the fiber in a transverse plane, $E^{m}$, $G^{m}$, and $\nu^{m}$ are Young’s modulus, shear modulus, and Poisson’s ratio of the matrix. In most cases, one can assume a value of 0.4 to 0.5 for *α*, and a value of 0.35 to 0.45 for *β* [21].

When the UD composite is only subjected to a transverse load *σ*_22_, the internal homogenized stresses in fiber and matrix are given as follows:(11)$$\sigma_{11}^{f}=-\frac{V_{m}A_{12}\sigma_{22}}{\left(V_{f}+V_{m}A_{11}\right)\left(V_{f}+V_{m}A_{22}\right)}$$
(12)$$\sigma_{11}^{m}=-\frac{V_{f}A_{12}\sigma_{22}}{\left(V_{f}+V_{m}A_{11}\right)\left(V_{f}+V_{m}A_{22}\right)}$$
(13)$$\sigma_{22}^{f}=\frac{\sigma_{22}}{V_{f}+V_{m}A_{22}},\;\sigma_{22}^{m}=\frac{A_{22}\sigma_{22}}{V_{f}+V_{m}A_{22}}$$
where $\sigma_{11}^{f}$ and $\sigma_{11}^{m}$ are the stress components of fiber and matrix along the longitudinal direction, while $\sigma_{22}^{f}$ and $\sigma_{22}^{m}$ are the stress components of fiber and matrix along the transverse direction. 

### 2.2. Stress Concentration Factor of Matrix under Transverse Loads

To determine the stress concentration factor of the matrix in a composite subjected to a specific load, pointwise stress in the matrix along the loading direction has to be obtained, which follows by choosing a suitable RVE for the composite. Although various RVE geometries are proposed in the literature [22], a simple concentric cylinder assemblage (CCA) model has been proven successful in a number of widely used micromechanical models. Hence, this work selected a CCA model as the RVE of a UD composite (as illustrated in Figure 1). 

When the CCA model is subjected to a transverse load $\sigma_{22}^{0}$, a pointwise stress component in the matrix along the loading direction ${\tilde{\sigma }}_{22}^{m}$ is given by Liu and Huang [16]
(14a)$${\tilde{\sigma }}_{22}^{m}={\tilde{\sigma }}_{\rho \rho }^{m}\cos^{2}\varphi +{\tilde{\sigma }}_{\varphi \varphi }^{m}\sin^{2}\varphi -{\tilde{\sigma }}_{\rho \varphi }^{m}\sin2\varphi$$
(14b)$${\tilde{\sigma }}_{\rho \rho }^{m}=\frac{\sigma_{22}^{0}}{2}\left\{1+Aa^{2}\rho^{-2}+\left[1+B\left(4a^{2}\rho^{-2}-3a^{4}\rho^{-4}\right)\right]\cos2\varphi \right\}$$
(14c)$${\tilde{\sigma }}_{\varphi \varphi }^{m}=\frac{\sigma_{22}^{0}}{2}\left[1-Aa^{2}\rho^{-2}-\left(1-3Ba^{4}\rho^{-4}\right)\cos2\varphi \right]$$
(14d)$${\tilde{\sigma }}_{\rho \varphi }^{m}=-\frac{\sigma_{22}^{0}}{2}\left[1-B\left(2a^{2}\rho^{-2}-3a^{4}\rho^{-4}\right)\right]\sin2\varphi$$
(14e)$$A=\frac{\left[1-\upsilon^{m}-2{\left(\upsilon^{m}\right)}^{2}\right]E_{22}^{f}-\left[1-\upsilon_{23}^{f}-2{\left(\upsilon_{23}^{f}\right)}^{2}\right]E^{m}}{\left(1+\upsilon^{m}\right)E_{22}^{f}+\left[1-\upsilon_{23}^{f}-2{\left(\upsilon_{23}^{f}\right)}^{2}\right]E^{m}}$$
(14f)$$B=\frac{\left(1+\upsilon_{23}^{f}\right)E^{m}-\left(1+\upsilon^{m}\right)E_{22}^{f}}{\left[\upsilon^{m}+4{\left(\upsilon^{m}\right)}^{2}-3\right]E_{22}^{f}-\left(1+\upsilon_{23}^{f}\right)E^{m}}$$

Note that according to the standard for the geometrical size of RVE, the diameters of the fiber and the matrix cylinders, 2*a* and 2*b*, are related to each other by the fiber volume fraction through
(15)$$b=\frac{a}{\sqrt{V_{f}}}$$

As discussed previously, the stress concentration factor of the matrix within a composite cannot be defined through a classical approach which divides maximum pointwise stress by the nominal stress. An improved approach [23,24] calculates the factor by using a line-averaged stress divided by a volume-averaged stress obtained from the previous bridging model. Figure 2 shows the schematic of the failure surface and the average line in the RVE of a UD composite subjected to a transverse tension load. In this configuration, the stress concentration factor of the matrix is given by
(16)$$K_{22}\left(\varphi \right)=\frac{1}{\left|{\vec{R}}_{\varphi }^{b}-{\vec{R}}_{\varphi }^{a}\right|}\int_{\left|{\vec{R}}_{\varphi }^{a}\right|}^{\left|{\vec{R}}_{\varphi }^{b}\right|}\frac{{\tilde{\sigma }}_{22}^{m}}{{\left(\sigma_{22}^{m}\right)}_{BM}}d\left|{\vec{R}}_{\varphi }\right|$$
where ${\tilde{\sigma }}_{22}^{m}$ is the pointwise stress of the matrix in a CCA model along the transverse direction, ${\left(\sigma_{22}^{m}\right)}_{\mathrm{BM}}$ is given by bridging model, *φ* is the inclined angle between the outward normal perpendicular to the failure surface and the loading direction denoted by *x*_2_, and ${\vec{R}}_{\varphi }^{b}$ and ${\vec{R}}_{\varphi }^{a}$ are respectively components of ${\vec{R}}_{\varphi }$ at the surfaces of the fiber and the matrix cylinders in the RVE.

In particular, ${\left(\sigma_{22}^{m}\right)}_{\mathrm{BM}}$ is the internal homogenized stress of the matrix in *x*_2_ direction, which is by nature an averaged quantity with respect to the matrix volume in the RVE. It can be computed as
(17)$${\left(\sigma_{22}^{m}\right)}_{BM}=\frac{\beta E_{22}^{f}+\left(1-\beta \right)E^{m}}{\left(V_{f}+V_{m}\beta \right)E_{22}^{f}+V_{m}\left(1-\beta \right)E^{m}}\sigma_{22}^{0}$$

Substituting Equations (14), (15), and (17) into Equation (16) gives an explicit form of Equation (15)
(18)$$\begin{array}{c} K_{22}\left(\varphi \right)=\left\{1+\frac{A}{2}\sqrt{V_{f}}\cos2\varphi +\frac{B}{2\left(1-\sqrt{V_{f}}\right)}\left[V_{f}^{2}\cos4\varphi +4V_{f}{\left(\cos\varphi \right)}^{2}\left(1-2\cos2\varphi \right)\right.\right.\\ \left.+\left.\sqrt{V_{f}}\left(2\cos2\varphi +\cos4\varphi \right)\right]\right\}\times \left[\frac{\left(V_{f}+V_{m}\beta \right)E_{22}^{f}+V_{m}\left(1-\beta \right)E^{m}}{\beta E_{22}^{f}+\left(1-\beta \right)E^{m}}\right] \end{array}$$

The only unknown parameter in Equation (18) is *φ*. Experimental observations have revealed that the failure surface is perpendicular to the loading direction when a UD composite is under transverse tension [24], in which case the inclined angle is zero.

Setting φ = 0 in Equation (18), the stress concentration factor is reduced to
(19)$$K_{22}^{t}=K_{22}\left(0\right)=\left[1+\frac{\sqrt{V_{f}}}{2}A+\frac{\sqrt{V_{f}}}{2}\left(3-V_{f}-\sqrt{V_{f}}\right)B\right]\times \left[\frac{\left(V_{f}+V_{m}\beta \right)E_{22}^{f}+V_{m}\left(1-\beta \right)E^{m}}{\beta E_{22}^{f}+\left(1-\beta \right)E^{m}}\right]$$

### 2.3. Failure Criteria for Constituents of a Composite under Transverse Tension

For a UD composite under transverse tensile load, the constituent fiber and matrix are both in a biaxial stressed state along the transverse and longitudinal directions, denoted by $\sigma_{22}^{f}$, $\sigma_{11}^{f}$, $\sigma_{22}^{m}$, and $\sigma_{11}^{m}$ in Equations (11)–(13). These orientations are also principal directions for the fiber and matrix. Moreover, it can be observed that the transverse stress components are the maximum principal stresses (note that $\sigma_{22}^{f}\gg \sigma_{11}^{f}$ and $\sigma_{22}^{m}\gg \sigma_{11}^{m}$). Due to the effect of stress concentration in the matrix, the homogenized matrix stress $\sigma_{22}^{m}$ in the transverse direction can be converted into the actual stress ${\bar{\sigma }}_{22}^{m}$
(20)$${\bar{\sigma }}_{22}^{m}=K_{22}^{t}\times \sigma_{22}^{m}$$

In addition, according to the explicit expression of fiber stress filed in CCA model [25,26], the pointwise stress of a fiber is uniform within it, thus the actual fiber stress in the transverse direction ${\bar{\sigma }}_{22}^{f}$ is equal to the homogenized value $\sigma_{22}^{f}$, i.e., ${\bar{\sigma }}_{22}^{f}=\sigma_{22}^{f}$. 

For a homogenized material under stress state where the maximum principal stress (minimum principal stress) is far greater (far smaller) than the other two quantities, the maximum normal stress theory is proven to be effective in predicting the ultimate strength. The stress status of fiber and matrix under transverse loads ($\sigma_{22}^{f}\gg \sigma_{11}^{f}$ and $\sigma_{22}^{m}\gg \sigma_{11}^{m}$) successfully satisfy the above use condition, thus, the strength theory of maximum normal stress will be used for the strength evaluation of fiber and matrix under transverse tension in this work. Specifically, when a UD composite fails under transverse tension, a combination of Equations (13) and (20) will result in the following relations
(21)$${\bar{\sigma }}_{22}^{f}=\sigma_{22}^{f}=\frac{\sigma_{22}^{u,t}}{V_{f}+V_{m}A_{22}}\geq \sigma_{u,t}^{f}$$
(22)$${\bar{\sigma }}_{22}^{m}=\frac{K_{22}^{t}A_{22}\sigma_{22}^{u,t}}{V_{f}+V_{m}A_{22}}\geq \sigma_{u,t}^{m}$$
where $\sigma_{22}^{u,t}$ is the ultimate strength of UD composites in transverse tensile direction, $\sigma_{u,t}^{f}$ and $\sigma_{u,t}^{m}$ are the ultimate strengths of fiber and matrix in tensile direction, respectively.

Using Equations (8), (21), and (22) can predict the transverse tensile strength of a UD composite as
(23)$$\sigma_{22}^{u,t}=\min\left\{\frac{\left(V_{f}+\beta V_{m}\right)E_{22}^{f}+\left(1-\beta \right)V_{m}E^{m}}{E_{22}^{f}}\sigma_{u,t}^{f},\frac{\left(V_{f}+\beta V_{m}\right)E_{22}^{f}+\left(1-\beta \right)V_{m}E^{m}}{\left[\beta E_{22}^{f}+\left(1-\beta \right)E^{m}\right]K_{22}^{t}}\sigma_{u,t}^{m}\right\}$$

### 2.4. Specimen Preparation

To verify the accuracy of the proposed micromechanics model, transverse tensile loading tests were carried out on a selected UD CFRP composite, namely CCF800H/AC531. Twenty layers of UD carbon fiber preform (CCF800H, Supplied by AVIC Composite Corporation Ltd., Beijing, China) were used to fabricate the desired laminates with constant fiber orientation (90°). Resin (AC531, supplied by AVIC Composite Corporation Ltd., Beijing, China) was used as matrix material. Required mechanical properties and the geometrical parameters of the constituent fiber CCF800H and matrix AC531 are provided in Table 1. After mixing these materials using a mechanical stirrer for 10 min, the mixture was kept in vacuum up to 15 min to ascertain a gas-free solution. Afterwards, a hand lay-up technique was used to apply the mixture on a carbon fiber preform in a desired sequence. A vacuum bagging method was employed to squeeze out the excess resin. The whole setup was then placed under 650 mm of Hg pressure for 24 h at room temperature. Subsequently, post-curing was executed at 80 °C for 8 h, placing the UD-CFRP laminates in a hot air oven. The fabricated laminate had a thickness of 2 mm with fiber volume fraction of 65%. Finally, according to nominal dimensions (Table 2) ruled by ASTM D3039 standard, six specimens were cut in the desired sizes as shown in Figure 3 using the abrasive waterjet cutting process.

### 2.5. Testing Methods

The transverse tensile loading test for the prepared composite specimens was conducted on a hydraulic mechanical testing system (MTS) by applying a controlled tensile load (Figure 4a). The test was performed based on the ASTM D3039 standard. The load history of the six samples was measured and recorded during the whole loading process. The load application on specimens was performed using a constant control rate of 2 mm/min. After performing the mechanical experiment, one of the fractured specimens was used for conducting an SEM test to further investigate the cause of the failure for the UD CFRP composite subjected to transverse tension. The SEM test was performed by JSM-F100 type, JEOL, Akishima, Japan (Figure 4b), for characterization of microcrack distribution inside the composite. The SEM sample, including the fracture surface, was acquired as a remaining piece with dimensions of 5 mm × 2 mm × 25 mm cut from the fractured specimen, which was subjected to transverse tension. Prior to the SEM test, a very thin carbon layer (1 nm) was used to sputter-coat the fracture surface [27].

## 3. Results and Discussion

### 3.1. Predicted Results

To make the predictions more illustrative, only the constituent elastic and the ultimate strength parameters in Table 1 were used, and no thermal residual stresses were taken into account. The bridging parameters *α* = 0.4, *β* = 0.4 in [*A_ij_*] were used. By substituting the above values into Equations (14e), (14f), (19), and (23), an ultimate strength of the UD composite CCF800H/AC531 and $K_{22}^{t}$ in transverse tensile direction were obtained as follows
(24)$$\begin{array}{l} K_{22}^{t}=2.92\\ \sigma_{22}^{u,t}=\min\left\{4540,58\right\}=58\;\mathrm{MPa} \end{array}$$

The factor $K_{22}^{t}$ has a value of 2.92, indicating the prominent effect of stress concentration on the matrix in transverse direction. It will significantly increase the actual stress in the matrix compared to the homogenized stress predicted by previous micromechanics models. The amplification of the matrix stress will in turn reduce the overall transverse tensile strength for the composite (in this case from expected value of over $\sigma_{u,t}^{m}$ (87 MPa) to an amended value of 58 MPa). This result demonstrates that the transverse failure of a UD composite is governed by matrix failure, completely in accordance with reported experiment findings.

### 3.2. Transverse Tensile Strength Test

The measured displacements against loads of the specimens are curved in Figure 5. Based on Figure 5 and the sizes of the specimen, transverse tensile strengths along with their relative errors (RE) with two groups of predicted results (bridging model + $K_{22}^{t}$ and bridging model) are listed in Table 3 and Figure 6. 

As seen from Table 3 and Figure 6, except for No.6, the measured transverse tensile strengths for all other five specimen show less than 10% RE from predicted strength by the proposed method (bridging model + $K_{22}^{t}$), whereas that for the conventional method (only bridging model) is almost 200%. The predicted outcome from the proposed micromechanics model is quite reasonable. Theoretically, the load-bearing capacity of UD composites in transverse direction is determined by the matrix. That is to say, the transverse tensile strength of a UD composite is at least as large as the matrix tensile strength. However, the range of measured transverse tensile strengths for the specimens is between 40 MPa–61 MPa, all much smaller than the measured pure matrix tensile strength (87 MP), with a maximum reduction of over 50%. The large deviation demonstrates that the effect of stress concentration on matrix can significantly weaken the transverse tensile strength of UD composites, so the present study on the calculation of a stress concentration factor is of great value for the safety of advanced composite material design. Additionally, as shown in Figure 7, it is observed that the fractured specimen exhibits failure surface perpendicular to the loading direction, which absolutely abides by the assumed angle φ = 0 in Section 2.2 to determine the stress concentration factor. 

### 3.3. Scanning Electron Microscope (SEM)

SEM analysis was performed to observe the initiation and evolution of microcracks inside the composite and explain the failure behavior of UD composite under transverse tensile loads. Figure 8 illustrates images of fracture topography for the sample. Figure 8a shows some microcracks initiation in the matrix oriented to fiber direction, while the fibers are almost undamaged. The typical length of the cracks is in micron scale, about 10 μm–30 μm. Given the fact that the strength of a matrix is much less than fiber and the effect of stress concentration on a matrix reduces its strength, the cracks are initiated first in the matrix along fiber direction for a UD composite under transverse tension. These cracks continue propagation slowly along the fiber direction to an approximate 10-micron scale by the increasing loads (Figure 8b), until an overall fracture of the composite. Thus, it is concluded from the mechanical experiment and SEM test that failure of UD composites subjected to a transverse tensile load is mostly dependent on failure of matrix, which is largely consistent with results obtained by previous research studies.

## 4. Conclusions

In this study, the effect of stress concentration on a constituent matrix in FRP composites under transverse tensile load was investigated to make an accurate ultimate strength prediction for composites. The primary outcomes of this study are summarized as follows:Instead of traditional definition by mechanics of fracture, using line-averaged stress divided by volume-averaged homogenized stress in the present study, an explicit expression for the stress concentration factor of the matrix in a UD composite subjected to a transverse tension was derived. With the addition of that factor, the stress state in matrix was revised as input data for failure criterion of composite.Following the results of conducted experiments on six 90° CFRP specimens in this study, the predicted transverse tensile strengths of the specimens agree well with measured results for an averaged error of 5.5%, while the error is over 200% for the conventional method which ignores the effect of stress concentration on a matrix. Thus, the proposed micromechanics method is feasible in predicting strength of a UD composite.The measured transverse tensile strengths of the specimens spread between 40–61 MPa, all much smaller than the pure matrix tensile strength (87 MP), in contrast with common knowledge that the transverse tensile strength of a UD composite should be at least as large as the matrix tensile strength. The reason for the strength reduction is the stress concentration in the matrix by incorporation of fiber into composite.The failure surface of specimen was perpendicular to the loading direction, indicating that the failure mechanism of the matrix under transverse tension follows the maximum normal stress theory.SEM images showed different scaled microcracks in the matrix oriented to fiber direction, while the fibers are almost undamaged. The cracks were initiated in micron scale, about 10–30 μm, and then propagated to a large scale by increasing loads until the overall fracture of the composite. The SEM analysis demonstrates that failure of UD composites subjected to a transverse tensile load is mostly dependent on a failure of the matrix.It is recommended to further investigate the effect of stress concentration on a matrix for UD composite under other loading conditions, such as transverse compression, longitude loads, and combined loads. Furthermore, plasticity of matrix could be considered in future studies in an effort to achieve more effective design of an advanced composite.

## Figures and Tables

**Figure 1 materials-15-08577-f001:**
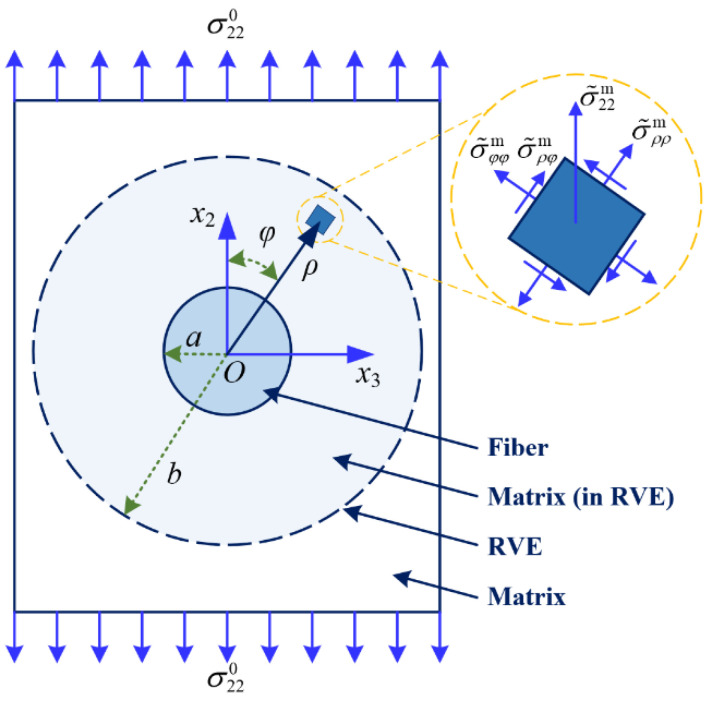
Illustration of the CCA model under transverse tension.

**Figure 2 materials-15-08577-f002:**
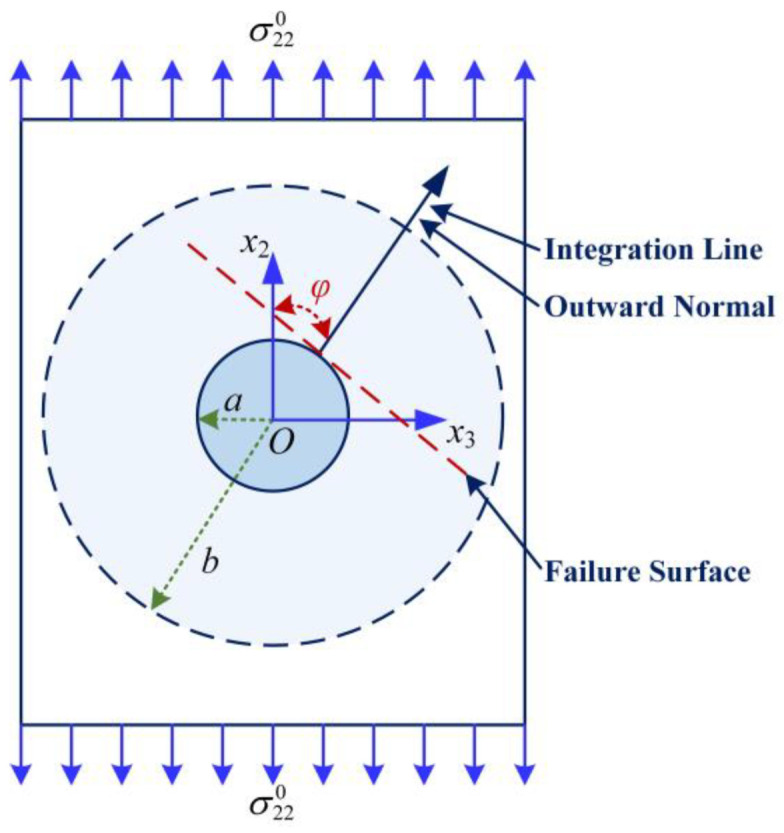
Schematic of the averaged stress used in defining stress concentration factor of the matrix in a composite subjected to a transverse load.

**Figure 3 materials-15-08577-f003:**
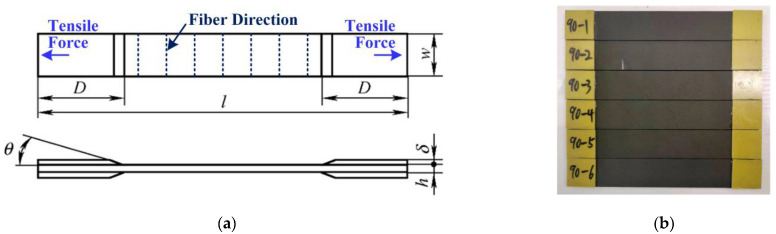
Specimen of 90° UD composite for transverse tensile loading test: (**a**) geometric design; (**b**) specimen photo.

**Figure 4 materials-15-08577-f004:**
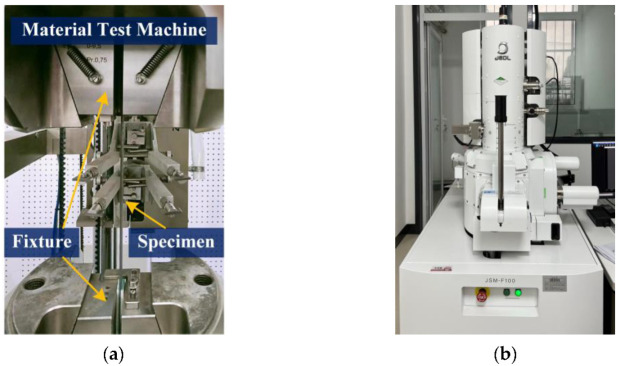
Devices for tests on the UD composite specimens: (**a**) setup for tensile test; (**b**) setup for SEM test.

**Figure 5 materials-15-08577-f005:**
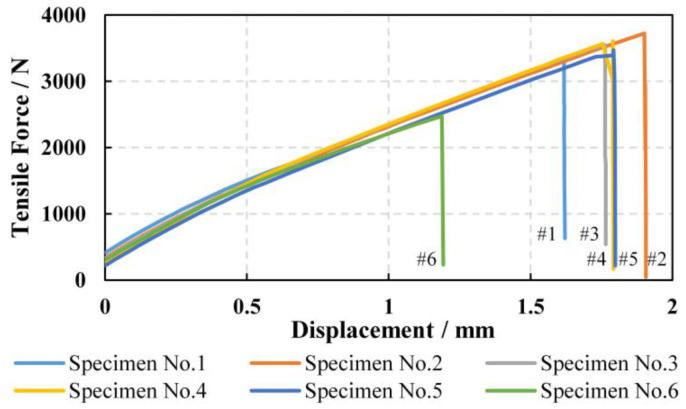
Displacement vs. transverse tensile force curves for six UD CFRP specimens.

**Figure 6 materials-15-08577-f006:**
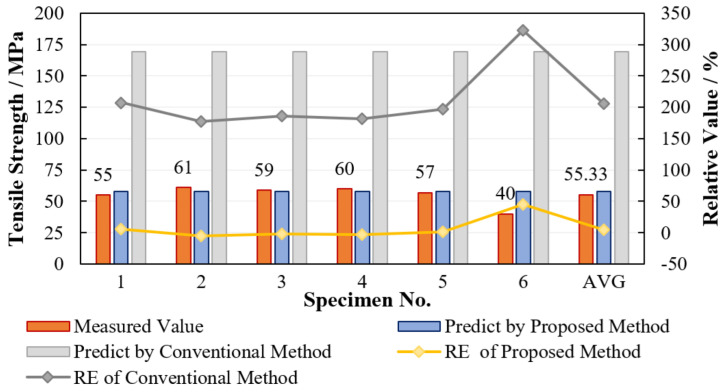
Comparison of predicted results with experimental results.

**Figure 7 materials-15-08577-f007:**
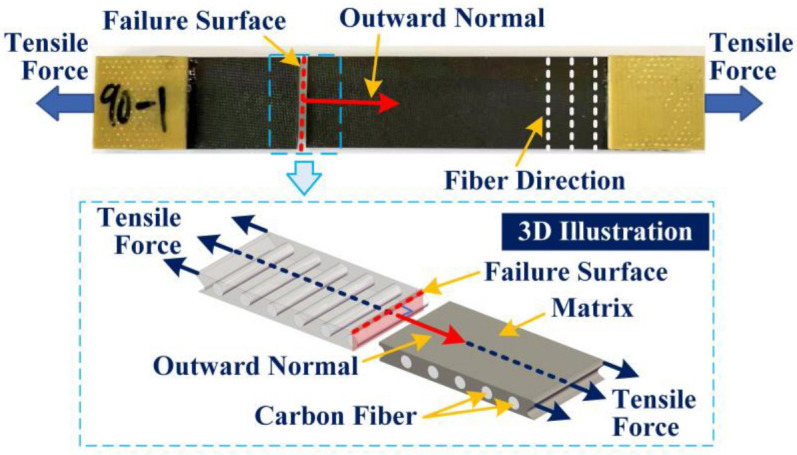
Fractured specimen illustration.

**Figure 8 materials-15-08577-f008:**
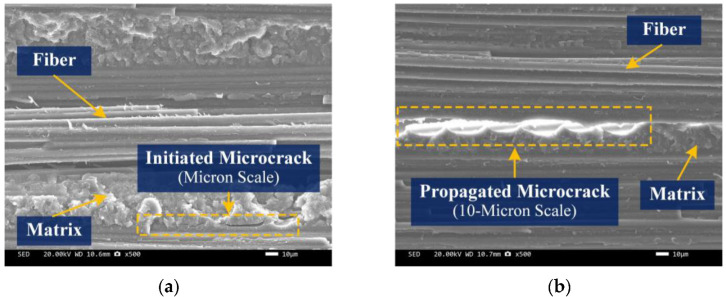
SEM images in different locations of the fracture cross-section of sample: (**a**) microcrack initiation; (**b**) microcrack propagation.

**Table 1 materials-15-08577-t001:** Constituent properties of CCF800H/AC531.

Fiber CCF800H	Property	Matrix AC531	Property
$E_{22}^{f}$ (GPa)	294	$E^{m}$ (GPa)	3.6
$\nu_{23}^{f}$	0.45	$\nu^{m}$	0.35
$\sigma_{u,t}^{f}$ (MPa)	5725	$\sigma_{u,t}^{m}$ (MPa)	87
*V* _f_	0.65	*V* _m_	0.35

**Table 2 materials-15-08577-t002:** Nominal values of geometrical parameters for the specimen.

Parameter	Value	Parameter	Value
*l*	175 mm	*δ*	1.5 mm
*W*	25 mm	*θ*	90°
*h*	2 mm	*V* _f_	0.65
*D*	25 mm	-	-

**Table 3 materials-15-08577-t003:** Transverse tensile strengths of the six UD composite specimens.

Specimen No.	1	2	3	4	5	6	AVG *
Pure Matrix Tensile Strength/MPa	87						
Predicted UD Transverse Tensile Strength/MPa (Bridging Model + $K_{22}^{t}$)	58						
Predicted UD Transverse Tensile Strength/MPa (Bridging Model)	169						
Measured UD Tensile Strength/MPa	55	61	59	60	57	40	55
Relative Error/% (Bridging Model + $K_{22}^{t}$)	5.45	−5.45	−1.69	−3.33	1.69	45	5.45
Relative error/% (Bridging Model)	207.27	177.05	186.44	181.67	196.49	322.50	207.27

***** AVG: average.

## Data Availability

Not applicable.

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
