# Peer review of "Micromechanics Modeling of Transverse Tensile Strength for Unidirectional CFRP Composite"

_materials, 2022, doi:10.3390/ma15238577_

Round 1
Reviewer 1 Report
Please further describe the main steps you followed and the main outstanding outcomes in the abstract.
Please further elaborate on the novelty of your work in the abstract.
Please include a brief but critical review regarding the conducted research studies in the introduction. It is recommended to add a section “research significance” and highlight the main contribution of your findings
Please include a brief summary of fiber reinforced polymers such as GFRP to improve structural responses of structures. Accordingly, please include the article titled Effect of Fiber Reinforced Polymer Tubes Filled with Recycled Materials and Concrete on Structural Capacity of Pile Foundations”.
Please explain the procedure of SEM images using the article titled Mechanical Characteristics of Cement Paste in the Presence of Carbon Nanotubes and Silica Oxide Nanoparticles: An Experimental Study.
Please include pictures of the specimens’ preparation steps.
Figure3: please add more explanation on the verification procedure of the conducted experiement. It is recommended to verify the performed tests by conducting finite element analyses.
Please add a comprehensive discussion on the main parameters that can significantly affect the reported outcomes in your work.
Please include a quantitative approach to report the test outcomes in the conclusion.
Please highlight the shortcomings in this research study and include recommendations for future research.
Reviewer 2 Report
The paper evaluated the effect of unidirectional FRP composite on the strength. The authors used Bridging Model for calculating the homogenized stress. The novelty is high. It can be suggested for publication after doing bellow corrections.
1- Please add a notation list.
2- Bellow papers investigated directional and unidirectional FRP composite. They should be reviewed and discussed in the Introduction Section.
SCFs in tubular X-joints retrofitted with FRP under out-of-plane bending moment. Marine Structures 79 (2021): 103010.
Multi-scale computational analysis of unidirectional carbon fiber reinforced polymer composites under various loading conditions. Composite Structures, 196, pp.30-43.
3- It seems, that you assumed perfect bonding between the FRP layers. There is not sliding between the layers? Please explain the reasons.
4- In figure 7b, the matrix can be seen in a limited zone. But, in figure 7a, it is propagated in all zones. What is the reasons. The reasons should be added in the paper.
5- Your conclusion is rather a summary. The conclusion and outcome is missing. How can the results be used? What remains to be researched?
6- What is the E1, E2, and E3 of the used FRP layers?
7- Minor corrections:
Line 82, “Huang [14] concluded “ is not correct. Please revise to “Huang and Xin [14] concluded “.
Line 84, “Gonzalez [16] reported” should be “Gonzalez and LLorca [16] reported”.
In Table 2, Whys is one value used for each parameter?
Line 115, “Equation (4) and (5) are” should be “Equations (4) and (5) are”.
Line 119, “β” should be Italic.
Line 135, “by [13]” should be “by Liu and Huang [13].”
Line 161, “Equation (14)-(15), (17)” should be “Equations (14), (15), and (17)
Line 232, please use one space in “Section3.2”.
Line 246, “Figure7(b))” should be “Figure 7(b))”
Reviewer 3 Report
Title: Micromechanics modeling of transverse tensile strength for 2 unidirectional CFRP composite
Reviewer comments:
This manuscript has been the focus on Micromechanics modeling of transverse tensile strength for 2 unidirectional CFRP composite. The study is original however I feel that the paper could be improved. Therefore, could you consider some points below for further improvement.
1. Abstract:
a) (Lines: 21 – 28): Please remove step by step operation method of system. Your abstract should be defining relevant parts of solution and results / implication.
2. Body:
a) (Lines: 98 – 99): Please redefine the equation (1) denotation, give a proper indication of the integration for the upper and lower limit.
b) Your writing is scatter with unorganized structural information. Please restructure your writing accordingly, follow the format IMRAD’s (Introduction, Material and methodology, Result and Discussion).
3. Results & discussion:
a) (Lines: 225 – 260): Table’s and Figure’s should place next to discussion paragraph, please re-arrange!!
b) (Lines: 257 – 258): Data from Table 3 is not complete; please fill up the data and complete it accordingly.
4. Conclusion:
a) The paper is a series of tests without explaining the outcome or the mechanistic reason behind the observations. Please highlight the significance of the overall result in terms of future application and methodology. Please improve!!
b) English need to be review.

Round 2
Reviewer 1 Report
N/A
Reviewer 2 Report
The paper suggests for publication.